# The Adjustment of Complexity on Sarcasm Processing in Chinese: Evidence from Reading Time Indicators

**DOI:** 10.3390/brainsci13020207

**Published:** 2023-01-26

**Authors:** Yutong Li, Hanwen Shi, Shan Li, Lei Gao, Xiaolei Gao

**Affiliations:** 1School of Psychology, Liaoning Normal University, Dalian 116029, China; 2School of Education, Tibet University, Lhasa 850000, China

**Keywords:** sarcasm, literal, text complexity, eye movement, Chinese, linear mixed model

## Abstract

It is controversial whether sarcasm processing should go through literal meaning processing. There is also a lack of eye movement evidence for Chinese sarcasm processing. In this study, we used eye movement experiments to explore the processing differences between sarcastic and literal meaning in Chinese text and whether this was regulated by sentence complexity. We manipulated the variables of complexity and literality. We recorded 33 participants’ eye movements when they were reading Chinese text and the results were analyzed by a linear mixed model. We found that, in the early stage of processing, there was no difference between the processing time of the sarcastic meaning and the literal meaning of simple remarks, whereas for complex remarks, the time needed to process the sarcastic meaning was longer than that needed to process the literal meaning. In the later stage of processing, regardless of complexity, the processing time of the sarcastic meaning was longer than that of the literal meaning. These results suggest that sarcastic speech processing in Chinese is influenced by literal meaning, and the effect of literal meaning on sarcastic remarks is regulated by complexity. Sarcastic meaning was expressed differently in different stages of processing. These results support the hierarchical salience hypothesis of the serial modular model.

## 1. Introduction

Irony is an indirect language and often signifies the opposite meaning of the literal meaning. Irony can be categorized as complimentary irony and critical irony. Critical irony is also called sarcasm, which expresses negative meanings in positive words [1]. In verbal communication, the use of sarcastic utterances is relatively common. In email communication with friends, 7.4% of people use sarcastic utterances, and about 8% of conversations between two friends contains sarcastic utterances [2,3].

The processing of sarcasm has attracted great attention from researchers. A central concern in the literature is the relationship between literal and sarcastic meanings. Which comes first: literality or sarcasm? There are two explanatory models: a serial modular model and a parallel interactive model. Serial modular models assume that sarcastic utterance processing is a sequential process, in which the literal meaning is always activated first. When the literal meaning is inconsistent with the context, the utterance processing continues until the sarcastic meaning is realized [4]. The most influential hypothesis of the serial modular model is the graded salience hypothesis. This hypothesis holds that linguistic meaning is not a simple dichotomy between literal meaning and sarcastic meaning but rather a continuum from extremely salient to non-salient. The salience of linguistic meaning is mainly determined by familiarity, convention, and typicality. In the cognitive processing of language, the more salient meaning is always activated first, followed by the less salient meaning [5,6,7]. In contrast to the serial modular model, parallel interactive models assume that both literal and sarcastic meanings are processed simultaneously. In the early stages of processing, literal and sarcastic meanings are extracted rapidly, and sarcastic meanings are selected in light of specific contexts [8]. To solve the problem of whether sarcastic speech processing is serial or parallel, researchers have conducted numerous studies.

In one study, researchers asked participants to read a passage of text that ended with a literal or sarcastic remark [9]. They found that there was no difference in the processing time between the literal and sarcastic remarks. This finding suggests that the sarcastic meaning can be accessed directly and not necessarily after the literal meaning is understood. In addition, Ivanko and Pexman used the moving-window paradigm to investigate the processing of literal and sarcastic meaning [10]. They also found that there was no time difference in the processing of literal and sarcastic meanings. These studies support the prediction of parallel interactive models. Dews and Winner used text dialogue materials in their research [11], where each dialogue contained a remark that was either literal or sarcastic. Participants were asked to make positive and negative judgments on the remarks after reading the material. The results showed that the time needed to judge the sarcastic meaning was longer than that needed to judge the literal meaning. Giora et al. explored sarcastic processing mechanisms using short dialogues [12]. The last sentence of every dialogue was a literal or sarcastic sentence. The results showed that the participants spent a longer time reading the sarcastic remarks than they needed to read the literal remarks. Moreover, the judgment time of the probing words related to sarcastic remarks was longer than that of the probing words related to literal remarks. These studies support the prediction of serial modular models. Sarcasm processing is influenced by many factors. One study found that contextual factors were the strongest predictor of perceived sarcasm [13]. Another study investigated the emotional function of irony by asking whether irony intensifies or mitigates negative feelings [14]. They found that whether irony intensifies or mitigates negative feelings depends on context.

So far, there is no consistent conclusion about sarcasm processing. These inconsistent results may be due to the different reading tasks and measures used in these studies [15]. For example, some studies have used a sarcastic evaluation task [11], whereas others use a lexical judgment task [10]. Some studies measure the processing time of the whole sentence [9], whereas others measure the processing time of individual words in a sentence [12]. Otherwise, the results of behavioral research have not been able to identify the specific stages in which the sarcastic and the literal meaning processing may be different. Some researchers are beginning to explore this question using eye-movement recording technology. Filik et al. asked participants to read text containing sarcastic sentences and recorded subjects’ eye movements [16]. Sarcastic sentences had different familiarity to the subjects. The results showed that there were no differences in gaze duration, regression path reading time, and total reading time between sarcastic remarks and literal remarks when the subject was familiar with sarcastic sentences. The sarcastic remarks required a longer gaze duration than the literal remarks when the subject was not familiar with the sarcastic sentences. This result indicated that the participants could detect the mismatch between sarcastic remarks and context at the early stage of processing unfamiliar sentences. Turcan and Filik manipulated the predictability of context so that the contextual expectation was either explicit or implicit to the participants [17]. When the subjects were reading, the researchers recorded their eye movements. They found that the processing of sarcastic meaning was longer than the processing of literal meaning in terms of first-pass reading time, regression-path reading time, and total reading time, regardless of whether the contextual expectation was implicit or explicit. These results suggested that processing of the literal meaning was faster than processing of the sarcastic meaning, and this processing pattern was not affected by context predictability. In addition, meta-analysis which analyzed the existing eye movement research on irony processing and the influencing factors of irony processing time showed that the most significant feature of irony processing was an increase in the re-read time of sarcastic words [18].

Turcan and Filik manipulated the antecedent conditions of context and recorded subjects’ eye movements while they were reading [19], and found that in terms of total reading time, processing literal meaning was faster than processing sarcastic meaning in the absence of antecedent conditions. Under antecedent conditions, however, they did not find any differences between sarcastic meaning and literal meaning processing. This finding suggested that antecedent conditions make sarcastic processing as easy as literal processing. Turcan et al. also manipulated speaker characteristics, which is a type of contextual information, in the experimental material and recorded subjects’ eye movements as they read the material [20]. They found that speaker characteristics significantly affected the processing of literal meaning and sarcastic meaning in terms of regression path reading time and total reading time. This study proved again that contextual information affects the processing of sarcastic remarks and supports the parallel interactive hypothesis.

The researchers also noted the impact of the complexity of the experimental materials on sarcasm processing. Some sarcasm is easy to understand, and the speaker’s intention can be explained by giving some words the opposite meaning of the literal meaning. However, some irony is difficult to understand, as the speaker’s intention cannot be obtained by changing the meaning of some words, and it requires a long chain of reasoning to understand [21]. Some researchers have evaluated the sarcasm, naturalness, and comprehensiveness of experimental materials, but have ignored the influence of complexity [20,22]. In some studies, the experimental materials include both simple and complex sentences [20] and in other studies, the experimental materials include only simple sentences [17]. When evaluating simple sarcasm, individuals can immediately detect a contrast between the meaning of the utterance and the context and can, therefore, understand the speaker’s intended use of sarcasm. When evaluating complex irony, individuals cannot immediately detect a contrast between the meaning of the utterance and the context, and they must understand the speaker’s ironic intention through complex reasoning. For ease of understanding, we cite examples from the review by Bosco and Bucciarelli [21]: Anita is with her friend Paolo and is looking for her glasses. She does not realize that her glasses are right in front of her nose. She asks Paolo, “Have you seen my glasses?”

Simple reply: “Congratulations on your excellent eyesight!”

Complex reply: “I would ask you if I had to thread a needle.”

Simple sarcasm requires individuals to use only simple reasoning to understand the true meaning of the utterance. The amount of time to process the literal meaning and to process the sarcastic meaning may not be different. In complex conditions, the processing time difference is greater between the literal meaning and the sarcastic meaning [21,23]. When comparing the processing differences between the literal meaning and sarcastic meaning without considering sentence complexity, it is impossible to judge whether the processing of sarcastic remarks is regulated by complexity.

Phonetic writing is the main language material used in prior research, and few studies have used ideograms such as those found in Chinese. Chinese is composed of Chinese characters, which are mainly ideographic. The composition of Chinese characters is very complex and there are no spaces between the words that make up Chinese sentences. In this study, we used Chinese texts to explore the processing patterns and time processes of Chinese sarcastic utterances with different degrees of complexity. We manipulated two independent variables: the literality (sarcastic, literal) and the complexity (simple, complex) of the remarks. Subjects were asked to read the material freely and to understand it as well as they could. We recorded subjects’ eye movements during reading. Referring to previous eye movement studies, we selected three indicators, namely, the first-pass reading time, the regression-path reading time, and the total reading time [19,20], because they represent the early and late stages of processing. These are reading time measures and measure the sum of eye fixation durations. Those measures summing temporally contiguous fixations can make an important contribution to the experimenters’ understanding of the precise pattern of eye movements [24]. There is a close correlation between the pattern of eye movements made by a reader and the mental processes needed to understand a text. The time taken to process the text is indicative of the ease with which processing occurred.

We hypothesized that sarcastic utterance processing is influenced by complexity. For simple sarcastic sentences, individuals directly access the sarcastic and literal meanings. For complex sarcastic sentences, however, individuals need to process the literal meaning first and then realize the sarcastic meaning. Specifically, in early eye movement indicators, we did not identify any differences in the processing time between sarcastic and literal meanings in simple sentences. In complex sentences, it took longer to process the sarcastic meaning than the literal meaning.

## 2. Materials and Methods

### 2.1. Participants

Thirty-three undergraduates (19 females; mean age = 23.1 years, *SD* = 2.2 years) participated in this experiment. All participants were native Chinese speakers, with no reading disorders and had normal or corrected-to-normal vision. The participants provided written informed consent and were financially compensated for their participation. The protocol of the experiment was reviewed and approved by the Institutional Review Board (or Ethics Committee) of Tibet University.

### 2.2. Materials

We created 64 experimental stimuli, each containing four sentences. The first sentence described the context in which the scenario was set. The second sentence described the development outcome of the event in the scenario, which was either good or bad. The third sentence was the character’s remark on the event, which was always a positive expression. We categorized remarks as simple or complex. The length of the remark was controlled (*M _simple_* = 7.13, *SD* = 0.81, *M _complex_* = 7.69, *SD* = 0.95, *t* (30) = −1.81, *p* = 0.08). The last sentence was a summation. Examples of the experimental materials are shown in Table 1.

#### 2.2.1. Complexity Pre-Test

We provided 18 native Chinese speakers (10 females; mean age = 25.1 years, *SD* = 2.3) with a questionnaire. This questionnaire contained context and remark sentences. They were asked to rate the complexity of each remark as sarcasm from 1 (simple) to 8 (complex). We presented participants with the complex and simple irony used by Bosco and Bucciarelli as examples, and gave the definition of complex evaluation, which requires a longer chain of reasoning [21]. When the subjects understood, they proceeded to conduct the complexity questionnaire of this study. We calculated the average score of each item. We then performed a *t*-test on the differences between the simple and complex groups. The results showed that the scores for complex remarks were significantly higher than the scores for simple remarks (*M _simple_* = 1.89, *SD* = 0.42, *M _complex_* = 5.31, *SD* = 0.57, *t* (30) = −19.22, *p* < 0.001).

#### 2.2.2. Familiarity Pre-Test

We provided 15 native Chinese speakers (8 females; mean age = 25 years, *SD* = 2.2) with a questionnaire comprising 32 remark sentences. Their task was to rate on a scale from 1 (unfamiliar) to 8 (familiar) their familiarity with the sarcastic meaning of each remark. This task is consistent with the task used by previous researchers in irony research [16,20]. We calculated the average score of each item. We then performed a t-test on the differences between the simple and complex groups. The results showed that there was no significant difference between the familiarity of simple remarks and complex remarks (*M _simple_* = 5.54, *SD* = 0.76, *M _complex_* = 5.02, *SD* = 0.92, *t* (30) = −1.76, *p* = 0.09).

Referring to previous studies, we created 64 fillers [17]. Half of the fillers had the same structure as the experimental stimulus, and the only difference between the filler material and experimental material was that the event outcomes in the filler materials were bad and the character’s remarks were criticisms. We included filler materials to prevent participants from expecting sarcastic remarks every time they read a negative scenario [25]. The other 32 fillers were introductions to scientific knowledge without any remarks. We did not perform data analysis on the filler materials.

Each scenario had four versions, and each version corresponded to an experimental condition. We set up four lists, and each list contained only one version of each scenario. Each participant was presented with one list to ensure that the participants were exposed to each scenario only once.

### 2.3. Design

The experiment was a 2 (complexity: simple vs. complex) × 2 (literality: literal vs. sarcastic) within-subject design. Thus, the experiment included four conditions: (1) simple literal remark (SL)—the result of the event in the second sentence is good, and the remark in the third sentence is simple; (2) complex literal remark (CL)—the result of the event in the second sentence is good, and the remark in the third sentence is complex; (3) simple sarcastic remark (SS)—the result of the event in the second sentence is bad, and the remark in the third sentence is simple; and (4) complex sarcastic remark (CS)—the result of the event in the second sentence is bad, and the remark in the third sentence is complex.

### 2.4. Apparatus and Procedure

We recorded eye movements using an SR Research Ltd. EyeLink 1000 eye-tracker (Ottawa, ON, Canada) that sampled eye position every millisecond. A chin rest and forehead rest minimized head movements. The eye tracker was calibrated using a 9-point horizontal calibration. Re-calibrations were undertaken if necessary. Materials were displayed on a computer screen at a distance of 56 cm from the participants’ eyes. The participants were instructed to read the materials as normally as possible for comprehension. Each trial consisted of one scenario, presented in its entirety on the screen, with two blank lines between each line of text.

When the subjects had finished reading and understood the text, they pressed the space bar to answer the question or to move on to the next trial. A quarter of the trials were followed by yes/no questions. The average correct rate of each participant was higher than 95%, which indicated that the participants had understood the materials.

### 2.5. Analysis

Many studies have reported typical word-level fixation time measures [26], which are not optimal for describing the time-course of sentence reading [27]. Sarcasm is a property of a phrase, not of a single word. Readers usually need to read an entire sentence to understand the sarcastic meaning of the sentence, and it is difficult to understand the meaning of a sarcastic sentence by reading only a single word. Therefore, the region of interest selected in this study was the entire sentence rather than a single word. We chose three interest regions. The first interest region was the second sentence in the scenario. The second interest region was the remark sentence in the scenario. The third interest region was the sentence following the remark sentence.

Eyetracking data were analyzed for fixations using the Eyelink DataViewer (SR Research, Hamilton, ON, Canada). We reported three measures of eye movement, first-pass reading time, regression-path reading time, and total reading time. First-pass reading time (fp) was the sum of all fixations in a region from first entering the region until after leaving it either through its left or right boundary. Regression-path reading time (rp) was the sum of all fixations in a region and in preceding regions from first entering the region to first going past it, that is, leaving it through its right boundary. Total reading time (tt) was the sum of all fixations in a region, including fixations made when re-reading the region. These three measures reflected the different stages of reading. First-pass reading time reflected the early processing stages and also showed the early difficulties that people may encounter when reading. Regression-path reading time included the first-pass reading time, plus any time spent re-reading earlier portions of the text to clarify difficult text. Total reading time was an indicator reflecting the late stage of reading processing, and it included both the first-pass reading time and any additional re-reading time.

We performed the data analysis in R [28] using linear mixed effects modeling (lme4 package). We reported the regression coefficients (*b*), *t*-values (*t*), *p*-values (*p*), 95% confidence intervals, and the random-effects structures with the variance and standard deviation (*SD*), where the lmerTest package was used to compute the *p*-values.

The first step was to establish the appropriate random-effects structure for each analysis. We started by fitting the maximal model to the data, as recommended by Barr et al. [29]. The random-effects structure of the maximal model was as follows: (1 + literality * complexity | subject) + (1 + literality * complexity | item). We used contrast coding to code the fixed effects: literal remark = 0, sarcastic remark = 1, simple remark = 0, complex remark = 1. We introduced literality and complexity as random slopes for both subjects and items because both factors were within-subject and within-item factors, respectively. Because the maximal model failed to converge, however, the random-effects structure had to be simplified to obtain convergence. We achieved convergence by progressively removing one random component at a time—the one that explained the least amount of variance in the previous nonconverging model.

Once we had established the random-effects structure, the next step was to perform a series of likelihood ratio tests comparing the fit of models with different fixed-effects structures to reach the best model fit for data. We compared the model with the two factors in an interaction with progressively simpler fixed-effects structures (i.e., two main effects but no interaction, or only one main effect). See Table 2 for the models that had the best fit for our data and the values of their fixed-effects parameters. Furthermore, see Table 3 for the series of likelihood ratio tests performed to reach the best models.

## 3. Results

### 3.1. The First Interest Region

First-pass reading time: No effects were observed in this index.

Regression-path reading time: No effects were observed in this index.

Total reading time: We observed a main effect of complexity (tt: *b* = −482.8, *t* = −3.3, *p* < 0.01, 2.5% CI = −774.4, 97.5% CI = −191.3). Compared with simple remarks, the total reading time for complex remarks was longer.

### 3.2. The Second Interest Region

First-pass reading time: We observed an interaction between literality and complexity (fp: *b* = −372.1, *t* = −4.1, *p* < 0.001, 2.5% CI = −551.4, 97.5% CI = −192.9). When the remark was complex, it took longer to read the sarcastic sentence than the literal sentence, but when the remark was simple, there was no difference in the reading time between the two sentences. When the remark was sarcastic, complex sentences triggered longer reading times than simple sentences, but when the remark was literal, there was no difference in reading time between the two sentences. We observed a main effect of literality in the first-pass reading time (fp: *b* = −359.2.1, *t* = 4.5, *p* < 0.001, 2.5% CI = −202.5, 97.5% CI = −512.9). Specifically, participants read literal remarks faster than they read sarcastic ones.

Regression-path reading time: We did not observe any effects of complexity and literality in this index.

Total reading time: We observed a main effect of complexity and a main effect of literality in the total reading time (tt literality: *b* = 383.9, *t* = 3.3, *p* < 0.01, 2.5% CI = 145.6, 97.5% CI = 622.0; tt complexity: *b* = −661.3.1, *t* = −5.2, *p* < 0.001, 2.5% CI = −921.7, 97.5% CI = −401.1). Specifically, the participants read literal remarks faster than they read sarcastic ones, and they read simple remarks faster than they read complex ones.

### 3.3. The Third Interest Region

First-pass reading time: We did not observe any effects of complexity and literality in the first-pass reading time.

Regression-path reading time: We observed a main effect of complexity and a main effect of literality in the regression path reading time (rp literality: *b* = 605.9, *t* = 2.1, *p* < 0.05, 2.5% CI = 30.7, 97.5% CI = 1181.2; rp complexity: *b* = −1006.1, *t* = −3.7, *p* < 0.01, 2.5% CI = −1556.3, 97.5% CI = −456.2). Specifically, the participants read literal remarks faster than they read sarcastic ones, and they read simple remarks faster than they read complex ones.

Total reading time: We observed a main effect of literality and complexity in the total reading time (tt literality: *b* = 141.1, *t* = 2.4, *p* < 0.05, 2.5% CI = 24.3, 97.5% CI = 257.9; tt complexity: *b* = −169.3, *t* = −2.3, *p* < 0.05, 2.5% CI = −320.8, 97.5% CI = −17.4). Specifically, the participants read literal remarks faster than they read sarcastic ones, and they read simple remarks faster than they read complex ones (see Figure 1 for reading time results).

## 4. Discussion

Eye movement results showed that the processing of sarcastic and literal meanings in Chinese text was moderated by the complexity of the remark, and this moderated effect was reflected mainly in the early stage of text processing. Individuals took significantly more time to process the sarcastic meaning than the literal meaning in the complex remark. We did not observe a time difference in processing the literal meaning and the sarcasm meaning in the simple remarks. In the integration stages of processing, the sarcastic meaning was processed more slowly than the literal meaning, regardless of the complexity. Moreover, this slowing down of processing existed in the spillover region. Generally speaking, it was more difficult to process the sarcastic meaning than the literal meaning, and complexity adjusted this processing.

In the first interest region, the effect of complexity appeared on the total reading time, and the reading time of complex remarks was significantly longer than that of simple remarks. At first glance, this result was difficult to understand because the complexity setting appeared after this region, and it did not follow the logic of cause and effect. In the process of reading, the previous content could be reprocessed by regression, resulting in the difference in the processing time of this region. Studies have shown that it is more difficult for individuals to understand complex speech than simple speech because understanding complex speech requires more complex reasoning processes [21,30,31]. The second interest region changed in complexity, resulting in more difficulty in complex remarks than in simple remarks. To better understand what was being read, readers look back at previous content. It can be seen that complex remarks increase the reprocessing time of the previous content. This finding fits with the natural reading process. This is also what studies using the sentence-by-sentence presentation paradigm cannot prove. 

In the second interest region, the most important result was the difference in the first-pass reading time. An interaction between complexity and literality appeared in this index. Specifically, the first-pass reading time of the sarcasm was significantly longer than that of the literal meaning in the complex remarks, whereas this difference disappeared in the simple remarks. Understanding complex evaluation is influenced by the complexity of the Chinese writing system. There are no spaces in Chinese sentences, and lexical segmentation is affected by the difficulty of sentence meaning. Turcan and Filik manipulated the familiarity of their remarks and found that if the remark was unfamiliar, the processing time of the sarcastic meaning was significantly longer than the literal meaning [19]. If the remarks were familiar, there was no difference in the processing time between sarcastic and literal remarks. Their results supported the graded salience hypothesis of the serial modular model. Although the salient meaning involves many aspects, researchers have often manipulated familiarity to change the salience of sentence meaning. The more familiar the sentence is, the more salient the semantic is [17]. A study by Filik et al. also found no difference in an early reading index between sarcastic and literal meanings when the remark was familiar because both were highly salient [16]. Their finding was consistent with the results of the present study. In the present study, the influence of familiarity was taken into account. Before the formal experiment, the familiarity of the experimental materials was controlled through a familiarity assessment. Thus, complexity was a major source of salience. For simple remarks, sarcastic meaning and literal meaning have a high degree of prominence, and the difference in processing time appeared only in the late stage. For complex remarks, the processing time of the sarcastic meaning was longer than that of the literal meaning. These results suggested that sarcastic utterance processing was moderated by sentence complexity. In this region, we also identified a main effect of literality on the first-pass reading time. The processing time of the literal meaning was shorter than that of the sarcastic meaning. This main effect should be interpreted with caution. The interaction of the first-pass reading time in this interest region showed that the processing time of the literal meaning was shorter than that of the sarcastic meaning in complex sentences. We can only guess that in complex sentences, it was more difficult to process the sarcastic meaning than the literal meaning. We cannot directly infer from the main effect that processing the sarcastic meaning was more difficult than processing the literal meaning. On the total reading time, we also identified the main effects of complexity and literality. Specifically, the processing time under complex remarks was longer than that under simple remarks. Previous studies have found that simple communication behaviors are easier to understand and faster to process than complex communication behaviors [32]. Similarly, simple remarks are easier to understand and faster to process than complex remarks. A main effect of literality has suggested that the processing time of literal meanings is shorter than that of sarcastic meanings. This may be because sarcasm processing requires more cognitive resources than literal processing in the integration stages of processing.

In the third interest region, the regression path reading time and the total reading time of complex remarks were longer than those of simple remarks. This result showed that compared with simple sentences, individuals had persistent difficulties in processing complex sentences. The regression-path reading time and the total reading time of sarcastic meaning were longer than those of literal meaning. This indicated that sarcastic meaning was more difficult to process than literal meaning in both simple and complex remarks. Turcan and Filik found that the processing time of sarcastic meaning was longer than that of literal meaning in the total reading time of the spillover region, which was consistent with the results of our study [17]. In the spillover region, the regression-path reading time and the total reading time of processing sarcastic meaning were longer than that of processing literal meaning. This showed that it was more difficult to integrate and process the sarcastic meaning than the literal meaning, and readers had to go back to the previous text to check and confirm the meaning. This study, however, did not calculate the index of readers reading the whole paragraph, which needs to be supported by more evidence.

In sum, sarcasm processing is more difficult than literal processing. This difficulty increases with the complexity of the language. For simple sentences with the same degree of prominence, the difficulty of sarcasm processing appears at the late stage of processing. The difficulty of sarcasm processing for complex sentences with different degrees of prominence always exists. Moreover, the difficulty of satirical processing leads to an increase in the review of the previous background information, which also affects the processing of the following content. The results of this study support the serial modular model. Literal meaning is always activated first, then sarcastic meaning processing begins.

This study had the following limitations. First, the experimental material is divided into four sentences. The background sentence is too simple and does not set off the background of the satire. Future research should design background sentences in detail. Second, although there was no difference between the familiarity of simple remarks and complex remarks (*p* = 0.09), a statistical trend in the matching of conditions is not ideal. The influence of this variable should be strictly controlled in future studies. Third, this study did not measure whether the participants really understood the material after reading it, nor did the study determine whether the individual’s eye movement results reflected the successful understanding of sarcastic speech or the failure to understand sarcastic speech. Forth, the control of the complexity of the experimental materials was categorized only into complex and simple, and the relationship between remarks with different levels of complexity and the sarcastic speech processing could not be explored in detail. This study found that the complexity of a remark could affect the time needed to process sarcastic language, and some studies have found that the familiarity of the attribute of a remark could also affect the time needed to process sarcastic language [17,19]. Whether other remark attributes affect the processing of sarcastic speech needs to be explored in future studies. In addition, the sentence reading process will also be affected by phonological processing. The differences in pronunciation between simple and complex remarks were not explored or controlled. Future studies should explore the interaction between the Chinese writing and pronunciation system. At last, only three eye movement metrics were analyzed in this study. In addition, conventional eye movement metrics include fixation count, saccade count, fixation frequency, saccade frequency, saccade average duration, and saccade total duration. Recently, new indexes have been proposed [33]: the fixation intersection coefficient, fixation intersection variability, fixation fractal dimension, active reading time, and saccade variability. The first three are spatial indicators and the last two are temporal indicators. More eye movement metrics can be used to answer the corresponding questions in more detail. Moreover, different types of eye movement metrics can solve the problem more comprehensively.

In general, this study manipulated the literality and complexity of remarks to explore the processing of sarcasm in Chinese. We found that in the early stages of reading, individuals processed the literal meaning of the sentence as quickly as the sarcastic meaning, and this process was moderated by complexity. When the remark became complex, the processing of sarcastic meaning was slower. These results not only support the graded salience hypothesis of the serial modular model but also extend the applicability of the hypothesis, i.e., this hypothesis applies not only to alphabetic characters but also to ideographic characters.

## 5. Conclusions

This eye movement study of Chinese sarcasm processing reveals that literality and complexity interact, and that the effect of literal meaning on sarcastic remarks is regulated by complexity. The difference in processing time between sarcastic meaning and literal meaning increases with complexity, and the more complex the sarcasm is, the more difficult it is to process. Early differences in visual processing are regulated by complexity. However, late visual processing differences remain. These results support the hierarchical salience hypothesis of the serial modular model. More research should be carried out to verify the processes of Chinese satirical processing, and the influence of complexity and familiarity on irony processing. In addition, the influence of speech processing should be explored.

## Figures and Tables

**Figure 1 brainsci-13-00207-f001:**
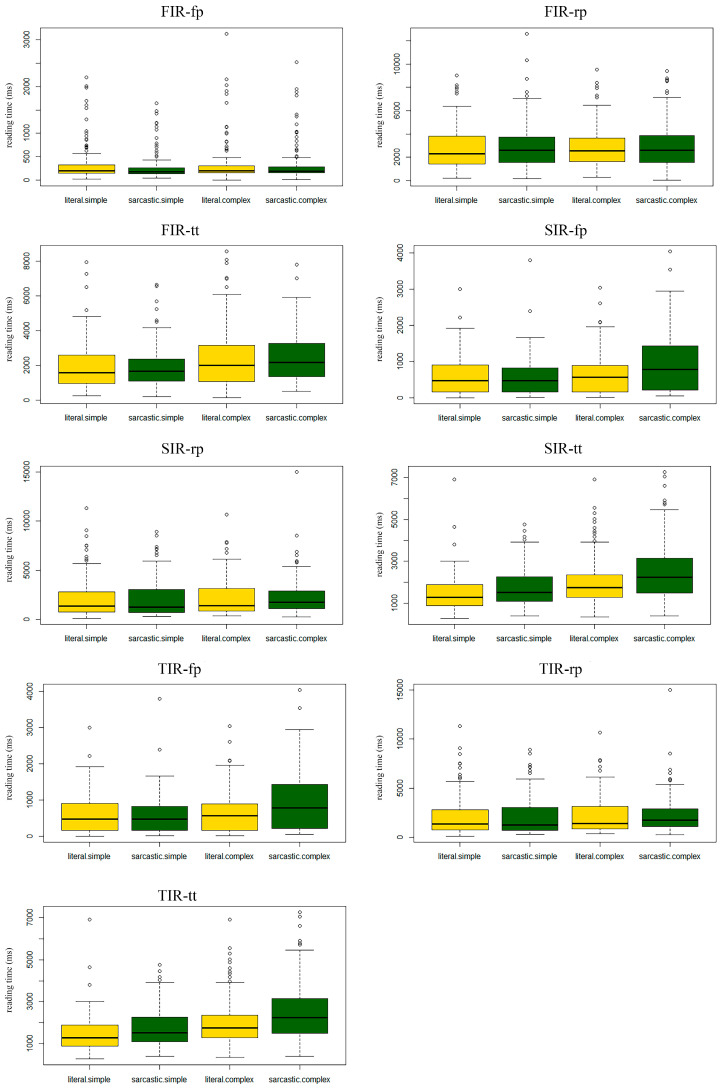
Reading times for the three regions of interest as a function of the type of sentence (literal, sarcastic) and level of complexity (simple, complex).

**Table 1 brainsci-13-00207-t001:** A sample set of experimental materials.

Lit	Com	Text (in Chinese)	Text (in English)
literal	simple	周末王涛和魏倩相约一起去看电影。 这个电影非常有趣。 魏倩说道：“这电影真好看”。 王涛说道：“我们等下去吃点夜宵吧”。	Wang Tao and Wei Qian met to watch a movie together on the weekend. This movie is very funny. “This movie is really good” Wei Qian said. “Let us eat supper” Wang Tao said.
	complex	周末王涛和魏倩相约一起去看电影。 这个电影非常有趣。 魏倩说道：“真是充实的两小时”。 王涛说道：“我们等下去吃点夜宵吧”。	Wang Tao and Wei Qian met to watch a movie together on the weekend. This movie is very funny. “It’s a fulfilling two hours” Wei Qian said. “Let us eat supper” Wang Tao said.
sarcastic	simple	周末王涛和魏倩相约一起去看电影。 这个电影非常无聊。 魏倩说道：“这电影真好看”。 王涛说道：“我们等下去吃点夜宵吧”。	Wang Tao and Wei Qian met to watch a movie together on the weekend. This movie is very boring. “This movie is really good” Wei Qian said. “Let us eat supper” Wang Tao said.
	complex	周末王涛和魏倩相约一起去看电影。 这个电影非常无聊。 魏倩说道：“真是充实的两小时”。 王涛说道：“我们等下去吃点夜宵吧”。	Wang Tao and Wei Qian met to watch a movie together on the weekend. This movie is very boring. “It’s a fulfilling two hours” Wei Qian said. “Let us eat supper” Wang Tao said.

Note: Lit-Literality: Com-Complexity.

**Table 2 brainsci-13-00207-t002:** Series of likelihood ratio tests, with their AIC and *p* values.

Model Number	Fixed-Effect Structure	fp		rp		tt	
		AIC	*p* (vs. Model Number)	AIC	*p* (vs. Model Number)	AIC	*p* (vs. Model Number)
First interest region							
1	lit * comp	7795		9092		8968	
2	lit + comp	7794	0.5 (vs.1)	9091	0.4 (vs.1)	8966	0.7 (vs.1)
3	lit	7792	0.4 (vs.2)	9089	0.6 (vs.2)	8973	<0.01 (vs.2)
4	comp	7794	0.2 (vs.2)	9089	0.5 (vs.2)	8965	0.8 (vs.2)
5	Intercept	7792	0.2 (vs.3) 0.4 (vs.4)	9088	0.5 (vs.3) 0.6 (vs.4)	8971	<0.01 (vs.4)
Second interest region							
1	lit * comp	8207		9415		8783	
2	lit + comp	8221	<0.001 (vs.1)	9413	0.8 (vs.1)	8784	0.1 (vs.1)
3	lit			9412	0.3 (vs.2)	8797	<0.001 (vs.2)
4	comp			9412	0.5 (vs.2)	8833	<0.01 (vs.2)
5	Intercept			9409	1 (vs.3) 1 (vs.4)		
Third interest region							
1	lit * comp	8132		9557		8411	
2	lit + comp	8131	0.2 (vs.1)	9555	0.9 (vs.1)	8410	0.6 (vs.1)
3	lit	8130	0.7 (vs.2)	9564	<0.01 (vs.2)	8412	<0.05 (vs.2)
4	comp	8130	0.5 (vs.2)	9558	<0.05 (vs.2)	8413	<0.05 (vs.2)
5	Intercept	8128	0.5 (vs.3) 0.7 (vs.4)				

Note: fp = first-pass; rp = regression-path reading time; tt = total reading time; lit = literality; comp = complexity. * represents all possible interaction items. The fixed-effect structure is simplified as the model number increases. AIC (Akaike’s information criterion).

**Table 3 brainsci-13-00207-t003:** Best fitting models and fixed-effects parameters and random structures.

AR	RM	Model	Fixed Effect	*b*	*t*	95% CI	
						2.5%	97.5%
FIR	fp	~1 + (1|subject) + (1|item)	(Intercept)	320.3	14.3 ***	276.2	364.5
	rp	~1 + (1 + lit|subject) + (1 + lit|item)	(Intercept)	2867.5	10.8 ***	2337.5	3398.8
	tt	~comp + (1 + lit|subject) + (1 + lit + comp|item)	(Intercept)	2441.6	10.5 ***	1977.4	2906.0
			comp	−482.8	−3.3 **	−774.4	−191.3
SIR	fp	~lit*comp + (1 + lit|subject) + (1|item)	(Intercept)	633.3	10.2 ***	511.0	755.6
			lit	359.2	4.5 ***	202.5	515.9
			comp	−43.4	−0.7	−170.0	83.5
			lit * comp	−372.1	−4.1 ***	−551.4	−192.9
	rp	~1 + (1 + comp|subject) + (1 + lit + comp|item)	(Intercept)	2259.4	11.8 ***	1875.9	2642.5
	tt	~lit + comp + (1|subject) + (1 + lit + comp|item)	(Intercept)	2083.8	13.6 ***	1780.3	2387.9
			lit	383.9	3.3 **	145.6	622.0
			comp	−661.3	−5.2 ***	−921.7	−401.1
TIR	fp	~1 + (1 + lit|subject) + (1 + comp|item)	(Intercept)	922.6	13.6 ***	787.0	1058.2
	rp	~lit + comp + (1 + comp|subject) + (1 + lit + comp|item)	(Intercept)	3567.7	9.8 ***	2848.0	4288.4
			lit	605.9	2.1 *	30.7	1181.2
			comp	−1006.1	−3.7 **	−1556.3	−456.2
	tt	~lit + comp + (1|subject) + (1 + comp|item)	(Intercept)	1477.2	13.3 ***	1257.9	1696.4
			lit	141.1	2.4 *	24.3	257.9
			comp	−169.3	−2.3 *	−320.8	−17.3

Notes. *** *p* ≤ 0.001; ** *p* ≤ 0.01; * *p* ≤ 0.05; CI = confidence interval; fp = first-pass; rp = regression-path reading time; tt = total reading time; lit = literality; comp = complexity; RM = reading measure; AR = analysis region; FIR = first interest region; SIR = second interest region; TIR = third interest region.

## Data Availability

Data materials can be obtained by contacting the corresponding author.

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
