# Peer review of "The Adjustment of Complexity on Sarcasm Processing in Chinese: Evidence from Reading Time Indicators"

_brainsci, 2023, doi:10.3390/brainsci13020207_

Round 1
Reviewer 1 Report
The paper is relatively interesting, but is a little 'messy'.
The Abstract starts too abruptly, does not present the research in itself, and proceeds by 'snaps', in a way which is not up to academic standards and not user-friendly for the audience (both specialized and non-specialized).
It needs to be re-written and, simply, 'normalized'.
The English language is clear enough, but, again, is not completely up to academic standards, it needs to be revised (thoroughly), with the help of a native-speaker.
The Introduction is ok, but it incorporates, like the following section on Materials and Methods, the literature review which, conversely, should be included in a specific section, entitled "Literature Review" and located between the Introduction and the Methodology sections. This would make the structure of the paper better and clearer and would help the readers, especially the non-specialized ones.
Moreover, the 'amount' of works used and cited in this paper is too 'little', the literature review in itself needs to be expanded, also by citing / commenting on works which are a bit more general and which can better show the position - and the relevance? - of this article in its field of studies.
The Methodology in itself (the "Methods", in the current second section) is too short, I mean, I get it, I understand it, it is, possibly, reproducible, but it needs to be expanded, at least for the sake of clarity, and enhanced, at least for the sake of comprehensiveness.
Results are ok, explained quite clearly, as it were, but at least the style of expression (the language) should be polished, in order to make their flow more 'fluid' (and 'fluent') - I am not sure to what extent the results are indicative, especially at the quantitative level, but this should be assessed by a specialist more specialized than me in the specific topic of the paper and, in a way, the evaluation is always 'subjective'.
The Discussion is respectable - it could be expanded still a little, with more comments and 'analytical moments', but nothing is ever perfect.
The Conclusions are laughable, truly. They need to be expanded considerably and, at least, they need to explain how the Authors have achieved their research goals and the significance of their paper in its specific panorama of studies. A Conclusion has to be a Conclusion, not two lines just to 'close' an article.
All in all, this paper has a lot of flaws, and I do not know to what extent it can be considered relevant, but I think it is readable and can help someone in a specific niche of studies - however, to do that, it needs to be thoroughly revised, and many issues have to be fixed, drastically. After that, it can be reconsidered for publication.
Thank you very much.
Reviewer 2 Report
I believe this study addresses an important issue. The research aim was very well established; methods were reasonably chosen and implemented with soundness. Results were presented in adequate detail, and interpreted with both insights and reference to existing literature. I have only two suggestions that may help further improve the readability and depth of the paper:
1. The procedure of the experiment can be further detailed, particularly the specific task processes of the experiment should be described;
2. Since the existing literature doesn't seem to lack studies on the same or a similar issue in alphabetic writing systems, the characteristics of the Chinese writing system needs to be further discussed as to their impacts on the experimental results. Meaning extraction in reading is not only a matter of decoding the text; it involves the auditory processing also. Then how a different type of auditory processing in native speakers of Chinese might influence the efficiency of meaning extraction may also be of some interest to the reader. At least this is worthwhile to be discussed towards the end of the discussion section.
Reviewer 3 Report
The paper titled: The adjustment of complexity on sarcasm processing in Chinese: Evidence from eye movement research presents an interesting study focusing on eye movement experiment, with the idea of exploring the processing difference between sarcastic and literal statements, and the influence of sentence complexity on sentence processing. The presented concept is interesting and does present interesting results and discussion, but the paper does have several aspects that should be improved before publication:
The abstract is heavily focused on discussing the obtained results but could benefit more from certain pieces of information:
a. The introduction of the hierarchical salience hypothesis in terms of sarcasm interpretation in the performed research.
b. the number of participants and their age
c. a mention of the method used for data analysis - linear mixed effects modeling
Keywords could be a lot more descriptive, consider adding more details such as: Chinese language, statistical analysis and text complexity (instead of just “complexity”)
There is an adequate ethics statement at the end of the paper, but the name of the committee should be added in the text as well, in the Participants section.
The title of the paper is slightly misleading. The phrase "Evidence from eye movement research" is not supported by the current methodology and results. The main methodological deficiency of this manuscript are rather basic eye movement measures utilized.
Authors chose reading time as the main measure of sarcasm processing, however the eye movement patters comprise more potential measures. Therefore, the more adequate title would be "Evidence from reading time indicators". Moreover, the introduction lacks the subsection on general relations between eye movements and measures of reading time (described in: Liversedge, S.P., Paterson, K.B. and Pickering, M.J., 1998. Eye movements and measures of reading time. In Eye guidance in reading and scene perception (pp. 55-75). Elsevier Science Ltd.)
In the introduction authors state only: "Referring to previous eye movement studies, we selected three indicators, namely, the first-pass reading time, the regression-path reading time, and the total reading time". Choosing a set of measures just by referring to previous work without a clear explanation of the underlying mechanisms supporting this choice is not adequate.
The Materials and methods section lacks a sub-section for data analysis (or a different suitable name). The description of the software used and the analysis procedure should not be stated in the results section (as they are currently) but should rather have their own subsection in the Materials and methods section. Furthermore, I understand that for the experiment design and extraction of the first-pass, regression-path and total reading time, dedicated software would have to be used. This software should be mentioned in the data analysis sub-section as well.
Certain lines in Table 2 seem to break in an awkward manner, for example, the first pass column, line 5 in the first interest region. Please correct this, as it would make the results easier to follow visually.
There is a typo in Table 3, in the AR column, the first stated value should be “FIR” as opposed to the current “IR”.
Figure 1 can help the reader visualize the differences between different scenarios but it is done quite poorly. Firstly, it should have a satisfactory resolution, as the letters are quite small and blurry. Please provide an image with a higher resolution, or one in a vector format (such as SVG or PDF). Secondly, the bar plots take up a lot of space (since they have a y-axis starting from zero) and contribute little to the visualization. I would strongly suggest using boxplots (or even violin plots) to visualize the data distributions. Finally, I believe that if boxplots are used, the y-axis values could be fixed across a single row (or column, or both) so that the results would be easier to compare. I also understand that the “***” denotes a p-value lower than 0.001 but this should be mentioned somewhere in the figure or the figure caption.
The limitations of the work are mentioned in the discussion, and certain ideas for future work are given, but I would like to suggest giving a short resume of the ideas for future work in the conclusion.
Reviewer 4 Report
As stated in the comments to the authors the study is well conducted, and the results and the paper are generally clear. I have tried to spell out clearly the points that in my opinion need revision.
General comment
This eye movement study examines the effect of sentence complexity on sarcasm perception. The research is well conducted, and the results and the paper are generally clear. There are several points in which the presentation can be improved, as detailed below.
Specific points
Line 34-35: “However, it is not clear how sarcastic and literal meanings are processed in Chinese sentences.”
Is the study focused on understanding the processes underlying sarcasm or in examining sarcasm in Chinese? I think there is some ambiguity in the presentation across the whole paper and an attempt should be made to make this point more clearly. My understanding is that the main hypotheses of the study, spelled out in lines 154-160, concern the general role of sentence complexity in the understanding of sarcasm. Further, in view of the lack of previous data on ideographic scripts, the authors want to also examine this aspect. Whatever their choice the authors should make this point more explicitly.
In this specific sentence, I would delete “in Chinese sentences” (with no loss of information) as the interest in the Chinese script is not anticipated or explained here. This point will become clearer in the par starting by line 144.
Introduction. The presentation is generally informative and clear. However, there seem to also be some repetitions and inconsistencies. For example, at lines 70-71, the authors state that “Filik and Moxey suggested that these inconsistent results may be due to the different reading tasks and measures used in these studies”. Later (lines 115-116), it is said “Although researchers have used a consistent experimental paradigm and task and have measured a relatively uniform reading index, results still have been inconsistent.” I would ask the authors to go over the introduction to check for smoothness and consistency in their arguments.
Lines 72 and 74. “For example, some studies used the sarcastic evaluation task [11], whereas others used the lexical judgment task [10]. Some studies measure the processing time…”
Here and elsewhere in the paper, the authors sometimes use the present and sometimes the past tense. I would ask them to check the text for consistency.
Line 98. “meta-analysis technology”.
I wonder whether meta-analysis can be defined as a technology. Maybe “methodology”?
Line 138. “Complex sarcasm, however, requires a more complex reasoning process”.
This definition seems partly circular. Please, check.
Line 198-200. “…there was no significant difference between the familiarity of simple remarks and complex remarks (M simple = 5.54, SD = 0.76, M complex = 5.02, SD = 0.92, t (30) = −1.76, p = 0.09).”
The presence of a statistical trend in the matching of conditions is not ideal. The authors may try to control statistically for this near-the-border difference in the stimuli or may indicate this as a limitation of the current study (specifying the importance of improving this matching in future studies). See below for further comments on the study limitations.
Figure 1 presents the results for all three regions of interest. In the text I would refer to the figure starting from the first region of interest (line 288 ?).
Figure 1 legend reads “Figure 1. the figure present reading time results of three regions.”
I would simplify some aspects of it and at the same time include more information. Something like: “Figure 1. Reading times for the three regions of interest as a function of the type of sentence (literal, sarcastic) and level of complexity (simple, complex)”.
Discussion
The first lines of the Discussion (322-327) merely repeat the method of the research and dilute the argument without much use. The section could start more effectively by “Eye movement results showed that the processing of sarcastic and literal meanings…”.
Lines 335-337. “Generally speaking, it was more difficult to process the sarcastic meaning of Chinese text than the literal meaning, and complexity adjusted this processing.”
Here, there is some ambiguity about whether the results concern sarcasm in general or the Chinese language. I would try to keep these two aspects more clearly separated.
One should also keep in mind that the study only used Chinese stimuli and speakers. So, one should be aware that any statement on this language is not in direct comparison with other languages.
Lines 338-401
A large part of the discussion follows an articulation in terms of the three regions of interest considered. Thus, reference to other studies in the literature occurs within each of these frames.
One wonders whether it would be possible to add at the end of this part a more general paragraph that compares the present results to the previous ones in the literature and provides the reader with a theoretical synthesis of the general meaning of the present findings.
Line 402 and following.
The authors spell out the study's limitations, which is a positive thing. However, some of the comments are quite broad. This is particularly true of the first one. Clearly, studying by means of eye movement technology, the perception of sarcasm in a naturalistic fashion is a long-reaching goal (given that it will ever be possible). At any rate, this limitation applies to all previous studies on the topic, and one wonders whether this argument is really useful for understanding the present research. The second and third comments are more focused and may be fine.
As stated above, I would add a sentence about the familiarity issue.
Round 2
Reviewer 1 Report
The paper has been revised thoroughly.
The work developed is appreciated.
If the Editors will consider it relevant enough, it can be considered for publication, now.
Thank you.
Regards.
Author Response
Response: Many thanks to the reviewer for your evaluation of the paper.

Reviewer 3 Report
The authors have succesfully answered to most of my comments. However, I also suggest considering or at least mentioning different eye movement measures in the discussion, aside from reading times only. Eye tracking technique offers variety of classical but also novel metrics related to eye-movement events which could possibly reflect cognitive processing more accurately.
Some of the novel metrics are summarized in this recent paper:
Vajs, I., Ković, V., Papić, T., Savić, A. M., & Janković, M. M. (2022). Spatiotemporal eye-tracking feature set for improved recognition of dyslexic reading patterns in children. Sensors, 22(13), 4900.
Author Response
Response to Reviewer 3 Comments
The authors have successfully answered to most of my comments. However, I also suggest considering or at least mentioning different eye movement measures in the discussion, aside from reading times only. Eye tracking technique offers variety of classical but also novel metrics related to eye-movement events which could possibly reflect cognitive processing more accurately.
Some of the novel metrics are summarized in this recent paper:
Vajs, I., Ković, V., Papić, T., Savić, A. M., & Janković, M. M. (2022). Spatiotemporal eye-tracking feature set for improved recognition of dyslexic reading patterns in children. Sensors, 22(13), 4900.
Response: Thank you very much for the reviewer's approval of our modification. Other eye movement measures were added to the discussion as suggested by the reviewers.
